

# Cloudy sky contributions to the direct aerosol effect

Gunnar Myhre[1], Bjørn H. Samset[1], Christian W. Mohr[1], Kari Alterskjær[1], Yves Balkanski[2], Nicholas Bellouin[3], Mian Chin[4], James Haywood[5,6], Øivind Hodnebrog[1], Stefan Kinne[7], Guangxing Lin[8,9], Marianne T. Lund[1], Joyce. E. Penner[10], Michael Schulz[11], Nick Schutgens[12], Ragnhild B. Skeie[1], Philip Stier[13], Toshihiko Takemura[14], Kai Zhang[15]

[1]CICERO Center for International Climate Research, Oslo, Norway.
[2]Laboratoire des Sciences du Climat et de l'Environnement, CEA-CNRS-UVSQ-UPSaclay, Gif-sur-Yvette, France.
[3]Department of Meteorology, University of Reading, Reading, RG6 6BB, UK.
[4]NASA Goddard Space Flight Center, Greenbelt, MD 20771 USA.
[5]College of Engineering, Mathematics and Physical Sciences, University of Exeter, Exeter, EX4 4QF, UK
[6]Earth System and Mitigation Science, Met Office Hadley Centre, Exeter, EX1 3PB, UK
[7]Max Plank Institute for Meteorology, Hamburg, Germany.
[8]University of Michigan, Ann Arbor, MI, USA.
[9]now at Atmospheric Sciences and Global Change Division, Pacific Northwest National Laboratory, Richland, WA, USA
[10]Department of Climate and Space Sciences and Engineering, University of Michigan, U. S. A
[11]Norwegian Meteorological Institute, Oslo, Norway.
[12]Earth Sciences, Faculty of Science, Vrije Universiteit Amsterdam, Amsterdam, Netherlands.
[13]Atmospheric, Oceanic and Planetary Physics, Department of Physics, University of Oxford, UK
[14]Research Institute for Applied Mechanics, Kyushu University, Fukuoka, Japan.
[15]Pacific Northwest National Laboratory, Richland, WA, USA

Corresponding author: Gunnar Myhre (gunnar.myhre@cicero.oslo.no)

**Abstract:** The radiative forcing of the aerosol-radiation interaction can be decomposed into clear sky and cloudy sky portions. Two sets of multi-model simulations within AeroCom, combined with observational methods, and the time evolution of aerosol emissions over the industrial era show that the contribution from cloudy sky regions is likely weak. A mean of the simulations considered is $0.01 \pm 0.1$ Wm$^{-2}$. Multivariate data analysis of results from AeroCom Phase II shows that many factors influence the strength of the cloudy sky contribution to the forcing of the aerosol-radiation interaction. Overall, single scattering albedo of anthropogenic aerosols and the interaction of aerosols with the shortwave cloud radiative effects are found to be important factors. A more dedicated focus on the contribution from the cloud free and cloud covered sky fraction respectively to the aerosol-radiation interaction will benefit the quantification of the radiative forcing and its uncertainty range.



# 1 Introduction

The radiative forcing (RF) of atmospheric aerosols due to the aerosol-radiation interaction – RFari (earlier denoted as direct aerosol effect) was assessed as -0.35 [-0.85 to +0.15] Wm$^{-2}$ in the Fifth Assessment Report by the Intergovernmental Panel on Climate Chang (IPCC) (AR5) (Boucher et al., 2013). The AR5 uncertainty range is even slightly wider than in the Fourth Assessment Report (Forster et al., 2007). Despite major progress in the understanding of atmospheric aerosol composition, and almost two decades of multi-aerosol type model simulations, little progress had been made in reducing the large uncertainty

in this number, until recently where Bellouin et al. estimate a range of -0.45 to -0.05 Wm$^{-2}$. Bellouin et al. estimate RFari from normalized clear sky radiative effect by aerosol optical depth (AOD) and multiply this by an assessment of anthropogenic AOD. No direct simulations were used to calculate the RFari in regions of clouds.

One reason for the larger uncertainty range in AR5 compared to AR4 was enhanced uncertainty and magnitude of the RFari of black carbon (BC) (Boucher et al., 2013). Bond et al. (2013) indicated that emission of BC was too low in the inventories

applied in climate models and therefore scaled the RFari from models to observed Aeronet absorption aerosol optical depth (AAOD) retrievals. More recently it has been suggested that AAOD data from Aeronet may have a sampling bias due to sites being located close to emission source regions (Wang et al., 2018), but uncertain in magnitude (Schutgens, 2019) and that most global aerosol models may have a bias towards too much BC in the middle and upper troposphere (Kipling et al., 2013; Samset et al., 2014; Wang et al., 2014). Both of these factors indicate a too strong BC RFari in Bond et al. (2013). However,

the most recent fossil and biofuel BC emission inventory (Hoesly et al., 2018) is much higher than used in previous global modelling (Lamarque et al., 2010). These new findings indicate that the BC RFari may be stronger than what was given in some of the multi-model global aerosol modelling exercises (Myhre et al., 2013a; Schulz et al., 2006), but likely weaker than estimated in Bond et al. (2013). The uncertainties in the RFari are also large for other aerosol species. For nitrate (Bian et al., 2017) the abundance is particularly uncertain, and for organic aerosols uncertainties are large both due to abundance (Tsigaridis

et al., 2014) and the optical properties, particularly for brown carbon (Samset et al., 2018).

A further complication when estimating RFari is the atmospheric mix of scattering and absorbing aerosols. Since RFari is dependent on aerosol optical properties and the underlying albedo (Haywood and Shine, 1997), it is therefore also very dependent on where the aerosols are located relative to clouds (Takemura et al., 2002). Absorbing aerosols above clouds have a strong positive RFari (Chand et al., 2009; Keil and Haywood, 2003) but it becomes considerably weaker if the aerosols are

located below clouds (Takemura et al., 2002). Scattering aerosols above or below clouds may enhance the reflection of solar radiation in conditions of thin clouds. The all sky RFari can be separated into contributions from clear and cloudy sky portions:

$$\text{RFari}_{\text{all sky}} = (1 - AC) * \text{RFari}_{\text{clear}} + AC * \text{RFari}_{\text{cloud}} \qquad (1)$$

AC is the cloud fraction, RFari$_{\text{clear}}$ and RFari$_{\text{cloud}}$ are the clear sky and cloudy sky RFari, respectively. All three variables vary as a function of longitude, latitude and time. The RFari is the initial perturbation to top of the atmosphere (TOA) radiative

fluxes (the instantaneous RF which for aerosol is very similar to RF). Rapid adjustments from absorption by aerosols mostly



BC, may alter atmospheric temperatures, water vapour and clouds. The sum of RFari and rapid adjustments is denoted effective radiative forcing (Boucher et al., 2013; Myhre et al., 2013b; Sherwood et al., 2015). The rapid adjustment of absorbing aerosols can be strong and may counteract the RFari substantial (Smith et al., 2018). Since RFari includes no rapid adjustments, AC is constant in Equation 1. Oikawa et al. (2013); Oikawa et al. (2018) provide estimates of all sky, clear sky and cloudy sky

radiative effect of aerosols in different regions based on satellite retrievals of clouds and aerosols. These studies further describe large differences resulting from whether aerosols are below or above clouds. Lacagnina et al. (2017); Zhang et al. (2016) found large regional variation in the radiative effect of aerosols above clouds. Note that the above-mentioned studies investigate the current, total aerosol abundance which consist of anthropogenic and natural aerosols, whereas in terms of RFari only the anthropogenic aerosols are considered. Matus et al. (2019) combined satellite derived aerosol radiative effect with model

simulation of anthropogenic aerosol, to make an estimate of RFari.

The aim of the present study is to provide insight into factors determining the contribution from cloudy sky regions to the RFari (second term on the right-hand side of equation 1) from combining global models and observations. We present estimates of this quantity from several global studies and we use multivariate data analysis to provide insight on the core factors causing the diversity among models.

**2 Methods**

**2.1 Global estimates of cloudy sky contribution to RFari**

The models, experiments and RFari from AeroCom Phase I and II are documented in detail by Schulz et al. (2006) and Myhre et al. (2013a), respectively. We also analyze the historical evolution of RFari due to anthropogenic aerosols using output from a series of OsloCTM3 simulations (Lund et al., 2018) with emissions from the Community Emission Data System (CEDS)

(Hoesly et al., 2018) inventory from year 1750 to 2014. The OsloCTM3 is a global 3-dimensional chemistry-transport model driven by 3-hourly meteorological forecast data.

The analysis is further supplemented by variables from Equation 1 extracted from Myhre (2009), who presented results from OsloCTM2 and an observational based method to explain the difference in all sky RFari between observational based and global aerosol model approaches. The model simulations were made to investigate several assumptions on aerosol optical

properties and impacts of assumptions related to missing data and change in industrial era aerosol concentration for the observational method. The Max Planck Aerosol Climatology version 2 (MACv2) method combines aerosol optical properties from Aeronet, surface albedo and clouds from ISCCP, vertical profiles of absorbing aerosols from the ECHAM-HAM model, with multi-model mean data on anthropogenic fraction from AeroCom Phase I (Kinne, 2019a; Kinne, 2019b).

The cloudy sky contribution to all sky RFari is calculated from daily or monthly diagnostics of allsky RFari, RFari$_{clear}$ and AC.

RF is taken at the top of the atmosphere and all estimates are from pre-industrial to present.





## 2.2 Multivariate data analysis

Multivariate data analysis in this study is based on results from a subset of the models participating in AeroCom Phase II (CAM5, GOCART, HadGEM2, IMPACT, INCA, ECHAM-HAM, OsloCTM2 and SPRINTARS). These eight models participated in AeroCom Phase II experiment (Myhre et al., 2013a) with no constrain on aerosol processes and in addition the
host model AeroCom exercise with fixed aerosol optical properties (Stier et al., 2013). From the latter FIX2 and FIX3 experiments can be used to retrieve the contributions from cloudy sky to RFari in two highly idealized aerosol radiative properties experiments. FIX2 is a purely scattering aerosol case and FIX3 is an absorbing aerosol case. The origin of the different variables derived from AeroCom phase II model simulations (Myhre et al., 2013a; Samset et al., 2013) can be found Table 1.

The data is analyzed using principal component analysis (PCA). Here, the variables that may influence the cloudy sky RFari contribution to all sky RFari are orthogonally transformed into linearly uncorrelated variables named principal components (PCs). The transformation is defined so that the first principal component (PC1) accounts for most of variance exhibited by the underlying variables. Each following PC in turn has the highest variance possible assuming that it is orthogonal to the previous PC, successively explaining less of the magnitude of cloudy sky RFari. Data is normalized prior to PCA to ensure
comparison of variance between variables. PCA results are usually plotted in a biplot, where only PC1 and PC2 (the second PC) are plotted on the x- and y-axis, respectively, since the two PCs explain most of the variance. In the biplot variables are projected as vectors along PC1 and PC2. The combined length and direction of the vector indicates the correlation the variable has with PC1 and PC2. Values range from -1 to 1 indicating negative to positive correlation with the PC. A value of 0 indicates no correlation with the PC. Since the projected vectors are directional it is possible to have high correlation with PC1 (values
~ -1 or ~ 1), and poor correlation with PC2 (value ~ 0), or *visa-versa*. Variables that point in the same direction are positively correlated with each other. Variables that point in the opposite direction of each other are negatively correlated. Variables that are perpendicular to each other are not correlated with each other or may be partially positively and partially negatively correlated. The missing data (see Table 1) for two of the models (LMDZ-INCA and ECHAM-HAM) are filled-in using regularized iterative PCA. This technique estimates the missing values, based on the correlation between the variables and the
principal components (Josse and Husson, 2012).

The contribution of cloudy sky to RFari ("Cloudy") (second term in Equation 1) is added as a supplementary variable in the PCA. This is to ensure that this dependent variable, the cloudy sky contribution, does not influence the projected correlations the independent variables have on each other. The same approach is applied to FIX2 and FIX3, as these variables are not independent, as they are composed of the many of the variables used in the analysis. In addition, linear regression correlation
coefficients are calculated between all the variables to assess the individual relationships. No data imputation is used for the one to one linear regression.



## 3 Results

### 3.1 Estimates of cloudy sky contributions to RFari

Figure 1 shows the all sky RFari due to anthropogenic aerosols separated into clear sky (first term on right-hand side in equation
1) and cloudy sky (second term on right-hand side in equation 1)  portion from AeroCom Phase II models (Myhre et al., 2013a).
The uncertainty ranges given in the figure are one standard deviation among the global aerosol models in AeroCom Phase II.
The figure shows two main results, that the cloudy sky RFari is weak and that the uncertainties in the contributions from cloudy
sky and clear sky are substantial with the latter somewhat larger in magnitude.

Figure 2 shows an example from OsloCTM3 (Lund et al., 2018) of the spatial distribution of various terms given in equation
1. In the lower row of Figure 2 the annual mean AC, $RFari_{clear}$ and $RFari_{cloud}$ are shown, with strong negative $RFari_{clear}$ over
most land areas except over regions of high surface albedo such as deserts. The $RFari_{cloud}$ is particularly positive over regions
of biomass burning aerosols overlying low level stratocumulus, but also over parts of high aerosol abundance over China. The
second row shows the two terms on the right hand in equation 1, namely the contribution from the cloud free and cloudy
regions to the all sky RFari. The contribution from the clear sky regions (first term on the right-hand side of equation 1) is
much weaker than $RFari_{clear}$ itself since cloud fraction is high in many of the regions of anthropogenic aerosols. While the
influence from cloud fraction on the cloudy sky contribution to all sky RFari relative to $RFari_{cloud}$ is weak over biomass burning
regions, it weakens relative to the negative values in $RFari_{cloud}$ over many areas in the northern hemisphere. In the top row the
RFari all sky is the sum of the contributions from clear and cloudy regions where their importance for the RFari varies
regionally.

Figure 3a-d shows estimates of the contribution from cloud sky to RFari from several studies: two are multi-model studies,
one combines model and observational based methods and one study investigates the time evolution using one model. The two
multi-model AeroCom studies (Myhre et al., 2013a; Schulz et al., 2006) show that the sign varies among the global aerosol
models and that two versions from one model changes sign between the two AeroCom phases (two versions of ECHAM-
HAM, UIOCTM versus OsloCTM2, and UMI versus IMPACT). The two model versions INCA and LSCE have positive
values in both AeroCom phases. SPRINTARS has the strongest positive (and overall strongest magnitude) of cloudy sky RFari
contribution to all sky RFari in both AeroCom phases. About half of the models shown in Figure 3a and 3b have provided
sufficient diagnostics to extract estimates from both AeroCom phases. The AeroCom PhaseII results will be further discussed
in section 3.2.

In Myhre (2009) several experiments were performed to explain that differences in RFari between observational based methods
and global aerosol models arise from a relatively larger change in absorbing aerosols over the industrial era than in the current
abundance of the absorbing aerosols. Whereas an observational method uses aerosol optical properties from measurements of
the present time of the combined natural and anthropogenic aerosols, the models simulate a relatively larger change in the
abundance of anthropogenic absorbing aerosols than assuming no change in the industrial era aerosol optical properties. Figure



3c shows the contribution of the cloudy sky to RFari from several of these experiments. The two experiments
MODIS(SCREEN) and MODIS use satellite retrievals of aerosol optical depth (AOD), current aerosol optical properties
retrieved (single scattering albedo and asymmetry factor) from AERONET, and a model estimate of the anthropogenic AOD.
The difference between these experiments is that the MODIS experiment uses model information over regions of missing
AOD from the satellite retrievals, while these regions are ignored in MODIS(SCREEN). The contribution of cloudy sky to all
sky RFari is similar in these two experiments. On the other hand, in the experiment MODIS(Model), changes in the aerosol
optical properties from pre-industrial to present causes the change in sign in the cloud sky contribution to RFari compared to
MODIS and MODIS(SCREEN). The MODIS(Model) has very similar RFari, as well as cloudy sky contribution to all sky
RFari, to the standard global aerosol model simulations (MODEL INT and MODEL EXT). The two latter model simulations
differ on whether internal or external mixing of BC is taken into account or not, respectively. The MACv2 cloudy sky
contribution to RFari is -0.13 Wm$^{-2}$. This estimate does not consider change in the aerosol optical properties over the industrial
era and can thus be compared to MODIS(SCREEN) and MODIS experiments described above.

The time evolution of the contribution of cloudy sky to RFari in OsloCTM3 is shown in Figure 3d where all variations are
caused by changes in the anthropogenic aerosol composition and abundance since all other factors are kept constant. Values
are negative in the period 1960 to 1990 due to a strong increase in SO$_2$ emissions and thereby a domination of scattering
aerosols and radiative impacts even in cloudy skies. In the period after 1990 the regional SO$_2$ emissions have changed strongly,
but with a small reduction in the global emissions. Emissions of BC have on the other hand increased substantially making
anthropogenic aerosols more absorbing in the OsloCTM3 causing a relatively stronger positive contribution from the cloudy
sky to RFari.

### 3.2 Multivariate data analysis of cloudy sky contribution to the all sky RFari

Table 1 lists the AeroCom Phase II models and variables included in the multivariate data analysis for investigating
contributions to cloudy sky RF denoted as "Cloudy" in the table (the second term on the right-hand side of equation 1). The
results of the multivariate data analysis are plotted in a biplot and a correlogram Figure 4. The principal component analysis
(PCA) found that 68.2% of the total variance is explained by the first and second principal component (PC1 and PC2), see
Figure 4a. The analysis shows that several factors are important for the contribution of cloudy sky to RFari ("Cloudy"). Among
all variables total short-wave cloud radiative effect (SW_CRF) is the most important. Single scattering albedo (SSA) being a
crucial variable for the anthropogenic aerosols may potentially be an important factor (a higher SSA is expected to give a more
negative cloudy sky forcing). However, independent correlations plotted in the correlogram (Figure 4b) suggests that the
cloudy sky contribution to RFari and SSA is weak (r = 0.17). In depth analysis of the linear correlation between cloudy sky to
RFari and SSA suggests that the linear relationship exist only at higher PCs (see supplementary Figure S1).



The contribution of cloudy sky to RFari shows a closer dependence on similar quantities for the idealized experiment FIX3
than FIX2, where FIX2 has purely scattering aerosols. Both FIX2 and FIX3 are dependent on host model properties such as
clouds, surface albedo and radiative transfer schemes (Stier et al., 2013).

PCA finds negative correlation between cloudy sky contribution to RFari and total short-wave cloud radiative flux (SW_CRF),
also supported by the linear regression. One example here is the GOCART model with the weakest SW_CRF and most negative
cloudy sky contribution to RFari of the model included in the multivariate data analysis. At the same time the PCA finds small
dependence between the cloudy sky contribution to RFari and cloud fraction (CLD_FR) or cloud altitude (CL_ALT). The
negative correlation between cloudy sky contribution to RFari and SW_CRF can be explained by reflective clouds enhancing
the underlying albedo and thus making the radiative forcing more positive with an increase in absorbing aerosols in the cloudy
sky portion.

Overall FIX3 (where models have a fixed SSA) and SW_CRF seem to be the main explanatory variables for the variance in
the cloudy sky contribution to RFari. It is however worth noting that the correlation for cloudy sky contribution to RFari with
the variables is not particular strong in any direction (indicated by the short arrow). This suggests that some of the variance
may be explained along the third or fourth principal component etc.

## 4 Discussion and conclusions

The multivariate data analysis shows that host model characteristics (especially SW_CRF) are important for the modeled
cloudy sky contribution to RFari, but also that many factors are important. Furthermore, several other studies presented here
show that aerosol properties (in particular SSA) are important for this quantity. Locally and especially in regions with aerosols
above clouds as well as in single model studies the SSA is crucial for cloudy sky contribution to RFari. However, analyzing
multi-model simulations then additional factors are becoming important. The two AeroCom phases give cloudy sky
contribution to RFari estimates of $0.0 \pm 0.10$ Wm$^{-2}$ and $0.04 \pm 0.10$ Wm$^{-2}$ and the mean of two observational based methods is
-0.02 (range from -0.13 to 0.09 Wm$^{-2}$). Combining the numbers from these three studies, we find $0.01 \pm 0.1$ Wm$^{-2}$ for the
cloudy sky contribution to all sky RFari. The new emission inventory from CEDS has a strong increase in BC emissions
leading to an increase in cloudy sky contribution to RFari of 0.05 Wm$^{-2}$ from 2000 to 2014 in OsloCTM3. Using OsloCTM3
simulations to investigate the importance of using diagnostics for every radiation time step (3 hourly) shows differences up to
0.01 W m$^{-2}$ relative to daily mean data and up to 0.04 W m$^{-2}$ for monthly data, but this may be model dependent (Haywood
and Shine, 1997).

Determining the quantity of black carbon from instrumentation such as the SP2 has provided a new set of consistent data for
assessing the performance of aerosol models (e.g. Kipling et al. (2013); Wang et al. (2014)). Knowledge of BC mass is
fundamentally insufficient for determining the ambient aerosol single scattering albedo owing to additional complexities such
as the degree of internal and external mixing. In the past, the aerosol modelling community has relied either on indirect





remotely sensed measurements from AERONET (e.g. Chin et al. (2009)) or on imperfect in-situ measurements of aerosol scattering from nephelometers (e.g. Anderson et al. (2003)) and absorption from filter-based systems (e.g. Bond et al. (1999)). Both of these systems are relatively imprecise corrections to account for scattering and absorption artifacts (e.g. Davies et al. (2019); Massoli et al. (2009)). The single scattering albedos can be determined much more accurately using combinations of cavity ring-down measurements of extinction (e.g. Lack et al. (2006)) and photoacoustic measurements of aerosol absorption

(e.g. Baynard et al. (2006)). These instruments are becoming more routine on aircraft equipped for making atmospheric measurements that can make highly accurate assessments of the aerosol single scattering albedo at above-cloud altitudes (e.g. Davies et al. (2019); Langridge et al. (2011)). These measurements will provide an invaluable additional source of data for model evaluation.

       Koffi et al. (2016); Koffi et al. (2012) show that global aerosol models generally tend to have an overabundance of

aerosols at higher altitude compared to satellite retrievals from CALIPSO and Samset et al. (2014) show that the AeroCom models overestimate BC at mid and high tropospheric altitudes compared to aircraft measurements. Too much BC above the clouds would overestimate the contribution of the cloudy sky to RFari. On the other hand, Peers et al. (2016) show that over the biomass burning region in south Africa most of the AeroCom models underestimate the AAOD over the stratocumulus layer, which would underestimate the contribution of cloudy sky to RFari.

In future studies of RFari, particular attention should be put on how global aerosol models simulate the location of aerosols in relation to clouds and how aerosol optical properties change with altitude in regions with high cloud cover compared to measurements in order to further constrain the spread in the modelled cloudy sky contribution. Nowhere is this high sensitivity more clearly demonstrated than over the SE Atlantic where biomass burning aerosols over-lie (and sometimes interact with) relatively bright stratocumulus clouds (e.g. Zuidema et al. (2016)). In addition to further analysis of aerosol RF

in cloudy sky regions, more emphasize should be devoted to quantifying the RFari in cloud free regions and its trend (Paulot et al., 2018), where the magnitude of forcing is larger than in cloudy regions.

**Acknowledgments, Samples, and Data**

All AeroCom data are available at the AeroCom server (https://aerocom.met.no/). The OsloCTM data will be made available through NIRD Research Data Archive. GM received funding from the Research Council of Norway through the SUPER (grant

250573). PS was supported by the European Research Council (ERC) project constRaining the EffeCts of Aerosols on Precipitation (RECAP) under the European Union's Horizon 2020 research and innovation programme with grant agreement No 724602.



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



**Tables**

**Table 1. Diagnostics from AeroCom Phase 2 (Myhre et al., 2013a; Samset et al., 2013; Stier et al., 2013) used in multivariate data analysis to investigate factors influencing the contribution of cloudy sky to RFari. The variable "Cloudy" is the contribution of cloudy sky to RFari (second term on the right-hand side of equation 1). FIX2scat and FIX3abs are the contributions from cloudy**
**sky to RFari in two highly idealized aerosol radiative properties experiments in Stier et al. (2013), where FIX2scat is a purely scattering aerosol case and FIX3abs is an absorbing aerosol case. The other variables, from AeroCom phase II model simulations, are total short-wave cloud radiative effect (SW_CRF), cloud fraction (CLD_FR), weighed cloud height (CL_ALT), weighted anthropogenic aerosol height (AER_ALT), single scattering albedo (SSA) of anthropogenic aerosols and fraction of anthropogenic BC mass above 5 km (BC_mass_5km). The variables Cloudy, FIX2, FIX3 and SW_CRF are given in Wm$^{-2}$, CLD_FR and BC**
**mass>5km in percent, with SSA unitless. AER_ALT and CL_ALT are given in hPa where the pressure levels are weighted by aerosol extinction and cloud fractions, respectively.**

| | | Host model dependences | | | | | Aerosol properties | | |
|---|---|---|---|---|---|---|---|---|---|
| Models | Cloudy | FIX2scat | FIX3abs | SW CRF | CLD FR | CL ALT | AER ALT | SSA | BC mass >5km |
| | W m-2 | W m-2 | W m-2 | W m-2 | % | hPa | hPa | 1 | % |
| CAM5 | 0.121 | -1.8 | 1.8 | -48.4 | 64 | 592 | 908 | 0.901 | 18.1 |
| GOCART | -0.114 | -1.6 | 1.2 | -21.8 | 58 | 520 | 867 | 0.937 | 27.1 |
| HadGEM2 | 0.0554 | -1.1 | 1.5 | -53.1 | 55 | 638 | 921 | 0.947 | 33.6 |
| IMPACT | 0.114 | -1.5 | 2.1 | -68.6 | 66 | 554 | 850 | 0.973 | 5.8 |
| LMDZ-INCA | 0.0756 | -0.8 | 2.5 | -53.1 | 47 | 585 | NA | 0.968 | 28.9 |
| ECHAM-HAM | -0.0242 | -1.7 | 1.1 | NA | 63 | NA | NA | 0.936 | 10.8 |
| OsloCTM2 | 0.0934 | -1.4 | 1.4 | -49.3 | 62 | 616 | 885 | 0.939 | 30.1 |
| SPRINTARS | 0.155 | -1.5 | 1.7 | -47.4 | 60 | 525 | 913 | 0.958 | 30.3 |




## Figures

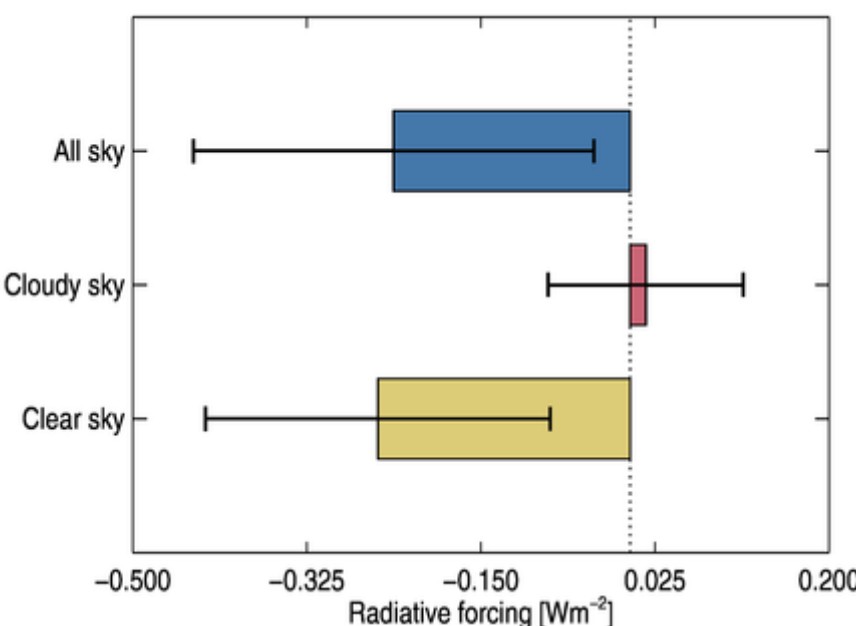


**Figure 1: All sky RFari, the contribution from cloud sky and clear sky to RFari (second and first term on the right hand of equation 1, respectively) from AeroCom Phase II simulations (Myhre et al., 2013a).**



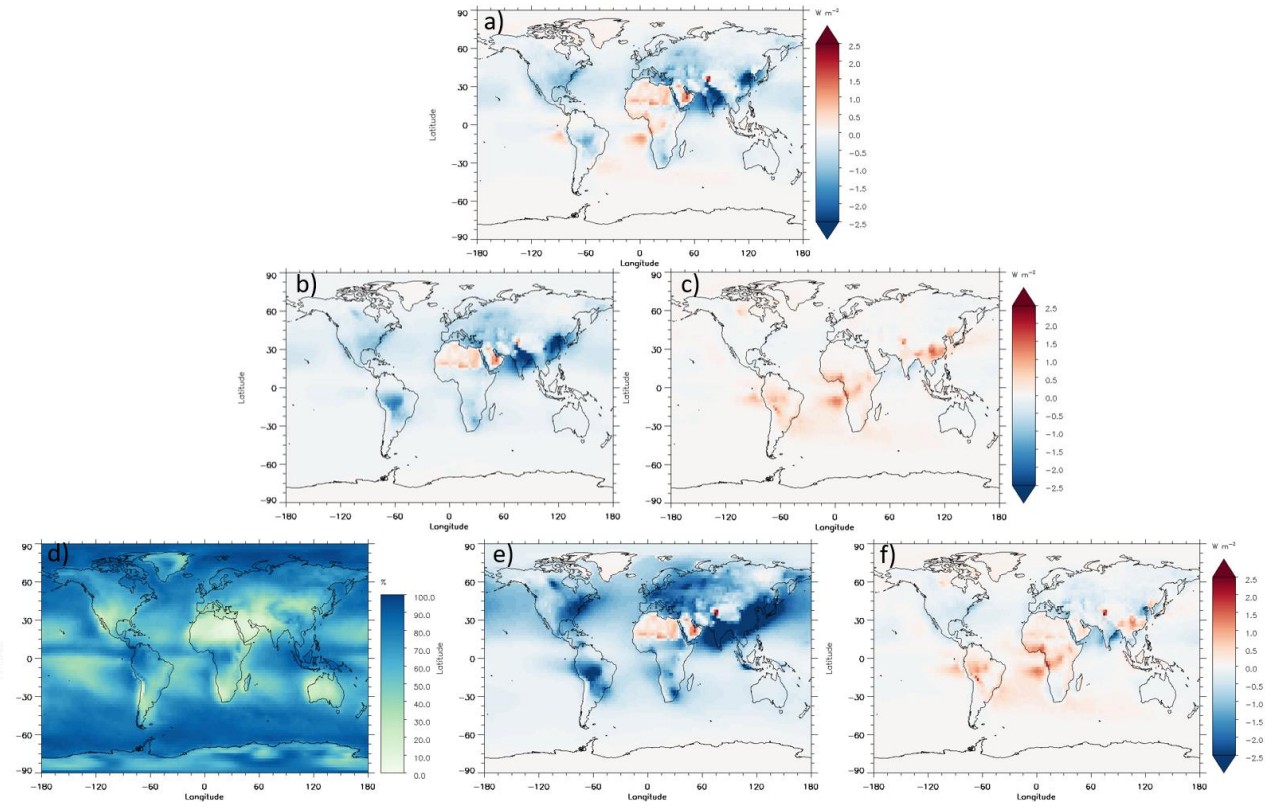

**Figure 2. Annual mean all sky RFari and various terms from clear and cloudy skies simulated with OsloCTM3 (Lund et al., 2018). The panel in the top row is the all sky RFari a), the second row shows the contributions from clear and cloudy sky (first b) and second term c) on right hand side on equation 1, respectively). The third row shows cloud fraction d), RFari$_{clear}$ e) and RFari$_{cloud}$, f) respectively (see equation 1). Panel d) on cloud fraction is showed in percent and the other panels in W m$^{-2}$.**







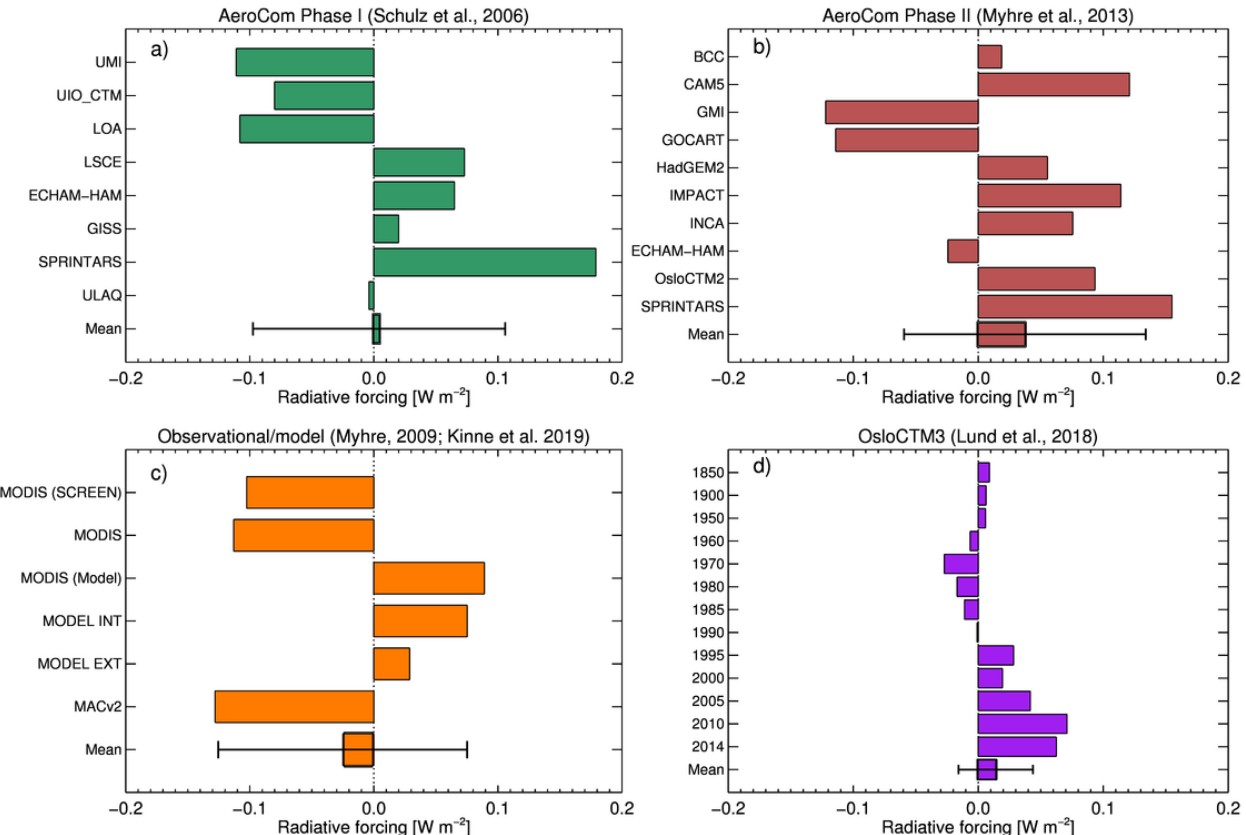

**Figure 3. The contribution of cloudy sky to RFari from AeroCom Phase I (Schulz et al., 2006) (a), AeroCom Phase II (Myhre et al., 2013a) (b), combination of observational based and model simulation (Kinne, 2019a; Myhre, 2009), where mean and standard deviation are based on all methods used in the panel (c), and time evolution from OsloCTM3 (Lund et al., 2018) (d).The multi-model mean is shown by the bars and the one standard uncertainty range of the models is given by whiskers.**




**(a)**

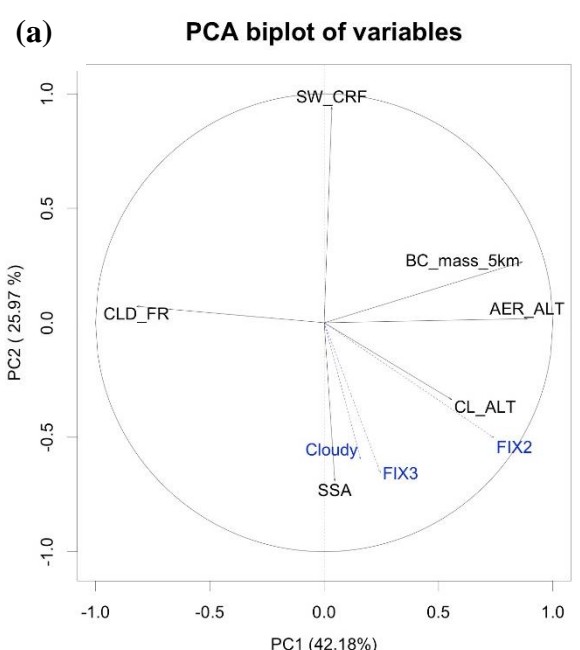

**(b)**

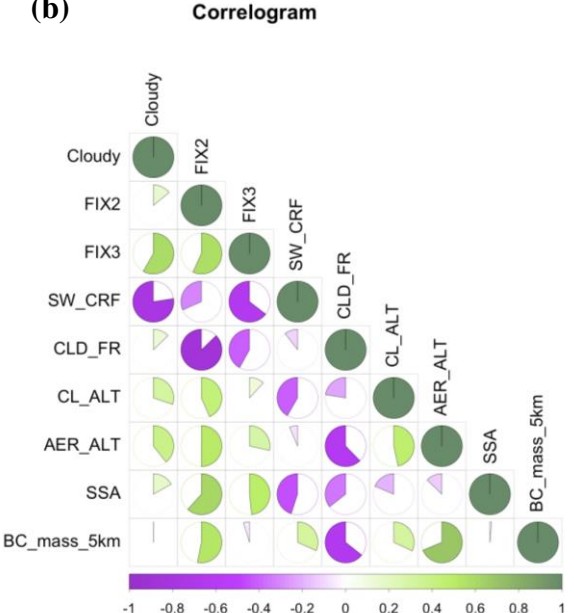

**(c)**

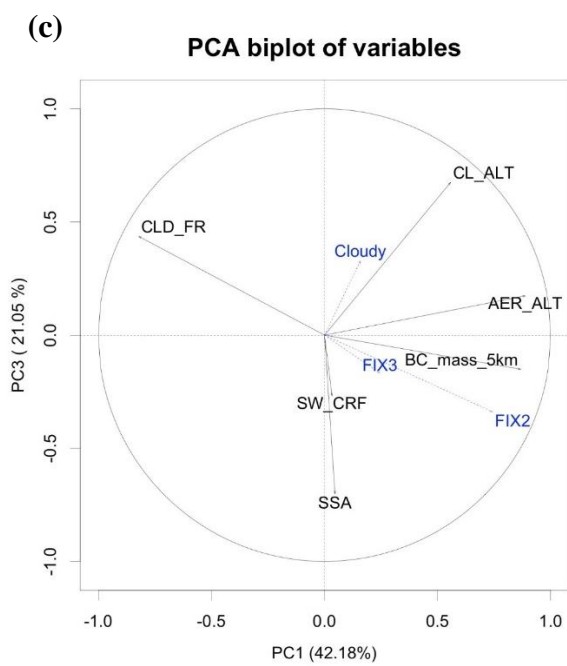





**Figure 4. Multivariate data analysis of eight AeroCom Phase II models (Myhre et al., 2013a) using diagnostics shown in Table S1.**
**Principal Component Analysis biplot of the variables (a). The length of the arrows indicates the strength of the correlation each**
**variable has in relation to PC1 and PC2, representing 42.2% and 26.0% of the variance respectively. Variables clustered together**
**indicate positive correlation with each other. Variable opposing each other indicate negative correlation. Cloudy, FIX3 and FIX2**
**(in blue) are added as supplementary variables, and do not influence the projection of the other variables. FIX2 and FIX3 have fixed**
**SSA globally and for all models, where the former experiment has pure scattering aerosol and FIX3 has relatively low SSA (and**
**thus high aerosol absorption). In (b), the correlogram shows the one to one linear regression correlation each variable has to each**
**other. The correlation coefficients (r) are presented on a color scale from -1 (purple) to 0 (white) to +1 (green). The strength of the**
**correlation is additionally presented as pie charts filling clockwise in green for positive correlations between two variables, and**
**counter clockwise in purple for negative correlation, where they can range from empty and full pie charts, indicating an absolute**
**correlation respectively from 0 to 1. Panel (c) is same as for (a), but for PC1 and PC3.**