# Peer review of "Cloudy sky contributions to the direct aerosol effect"

_Atmospheric Chemistry and Physics, 2019_

## Referee Comment (RC1) · Anonymous Referee #1 · 19 Jan 2020

General

The paper presents a modelling study on the direct aerosol effect on climate. The authors distinguish between clear and cloudy skies. The approach is probably state of the art although, however, to my opinion, a very simple one.

Let me start with my general impression: We have satellite lidars delivering global 3-D aerosol distributions (profiles!) with detailed aerosol typing (in terms of optical, microphysical and even chemical composition and thus refractive index characteristics) around the globe from the surface up to stratospheric heights and also producing 3-D distributions of clouds layers, their thermodynamic phase, frequency and cloud cover. In addition, we have sophisticated passive remote sensing techniques, again, delivering very detailed information on cloud layering, cloud heights, cloud types, cloud cover, and thermodynamic phase. In view of all the available and complex global 3-D cloud

and aerosol data sets, I am a bit surprized that teams of modellers still use rather simple approaches (here Eq.(1)) to investigate and estimate the role of aerosols (natural and anthropogenic ones) in the climate system with the goal to answer the very important and 'ultimate' question: What is the contribution of anthropogenic aerosols to climate change? Even if global MODIS column information on AOD (and maybe cloud occurrence and cover?) is included in the study, . . . is that sufficient to obtain a realistic picture on aerosol effects on climate? The global aerosol distribution (profiles) used in this manuscript is rather simple so that question arises: Does the modelled global aerosol climatology really reflect the real world?

Maybe, there are meanwhile modelling groups and thus papers in which the measured global aerosol distributions and measured global cloud distributions are used to model the impact of aerosols on global climate conditions, and these authors here just want to offer an alternative way, a more simple, rather basic approach to estimate the aerosol effects on climate? Maybe that is the reason for this simple paper but at the end the main question is still: Can we believe in these results when such a simple approach is used?

And, are you sure that you cover the full spectrum of anthropogenically caused aerosols. What about all the dust in the atmosphere especially over Central and East Asia, is that all natural? Clearly: NO! But how to consider that in the model? Did you consider that in the simulations? Probably not!

The paper is worth to be published, no doubt! The list of authors is full of well-known experts, and the paper is a valuable contribution to the climate debate, but the authors should at least try to provide some answers to my concerns. Yes, maybe I am 'naive' . . . as an experimentally working specialist for aerosol and cloud profiling, and my comments indicate that I am not familiar with the modern modelling world but I am probably not the only one who has trouble with the concept and content of this paper. Maybe, I completely missed the point and the overall message of the paper, but again, I will be probably not the only one. So, we need a more critical discussion on the

approach itself in this paper.

Details:

P2, l40: Bellouin et al. . ... this is obviously not a publication, there is no year of publication, nothing. So, that is not an acceptable statement. Please improve!

P2, l50: . . . biofuel BC emission inventory is much higher than used in previous global modelling . . .. Bad wording? What do you want to say?

P2, l62: Eq (1) is the most basic (trivial) approach, right? Or is there even a more simple one? On the other hand, the atmospheric system is so complex, and modern instrumentation fill the aerosol and cloud data base since 20 years, continuously. You seem to ignore all this! You separate (anthropogenic) aerosol particles in absorbing and non-absorbing ones, nothing else. Is that sufficient? You introduce AC as cloud fraction! Obviously it doesn't matter whether we have one layer, two layers, three layers of clouds, whether we have liquid-water clouds, mixed-phase clouds, cirrus . . .or even complex cloud mixtures and layering, and it is also not essential whether the aerosol is below the lowest cloud layer, between the different cloud layers, etc. . . Just one parameter is sufficient: AC! For the entire globe! For rather different climate zones? One AC value everywhere. . .? This is quiet a surprizing and 'universal' assumption. The other way around, what did I miss here? Please clarify, other readers (not familiar with climate modelling) may think the same. . ., may have the same problem with the paper. Maybe all the referenced papers show that it is sufficient to have just AC to describe the impact of clouds on the aerosol radiative effect around the globe from the tropics to the poles.

P3, l70: aerosols above clouds, below clouds. . . Only these two scenarios, not more are need to be modelled and considered? . . . although the world is full of complex aerosol and cloud layering. . . and large areas over the oceans downwind of polluting continents in the northern hemisphere . . . are 'affected' by this complex layering?

P3,l93: When using Stefan Kinne's aerosol climatology, did you at least check how good the agreement between CALIPSO aerosol profile observations (in combination with MERRA and CAMS simulations) and Kinne's aerosol climatology is? I speculate: Yes, you did that! My 'spontaneous feeling' is that this quiet simple aerosol profile climatology is not in good agreement with the real world. So, please comment on this!

I would suggest to include a figure with a sketch of your basic aerosol-cloud scenarios considered in the model. Show a cloud layer (provide information on the cloud height, then visualize AC, that means, the cloud should not cover the full sketch from left to right, and then indicate aerosols (just a mixture of black (absorbing) and yellow or white points (non absorbing particles). Scene 1: aerosol below the cloud, Scene 2: aerosol above the cloud layer, Scene 3: aerosol in the clear part of the sketch, if there are more scenes in the model, please continue with further scenes....

P5, l127: Result section: My only one question ... throughout this section... was at what height is the cloud layer (for which we have a fixed, constant AC)? Obviously you only consider liquid-water clouds in the lower troposphere. A cloud layer at, e.g., 1 km height (boundary layer top) almost everywhere.... around the globe. Maybe it is stated somewhere and I missed it unfortunately. But what about the impact of all the midlevel cloud fields (partly glaciated...) and the extended subvisible cirrus fields around the globe..., no impact on the aerosol related radiavtive effects?

The rest of the paper sounds ok (consistent) .... for a non-modelling atmospheric scientists traveling around the globe and measuring the rather complex world of clouds and aerosols in regions with very high amounts of haze and dust (which is partly triggered by human activities) and partly complex aerosol layering up to the tropopause, ... and, in contrast, in very pristine areas with simple cloud and aerosol layering as in your model.

My 'basic' comments may be confusing but the goal is to improve the paper, not to destroy it.

---

## Referee Comment (RC2) · Anonymous Referee #2 · 2 Mar 2020

In this work, the authors investigate the contribution of aerosol in cloudy skies to the magnitude of the aerosol direct effect (RFari). They results from a collection of global models to show that the contribution to the RFari from cloud skies is small. They also investigate the parameters that affect this between different models, showing that the shortwave cloud radiative effect is the biggest contributor to inter-model differences.

This work is within scope for ACP and would be relevant for the readers. I have some comments, particularly regarding the notation and some of the explanation, after which I believe it would be suitable for publication.

**Major points**

Has the choice for the meaning of RFari$_{clear}$ and RFari$_{cloudy}$ been made in a previous paper? If not, a change in the notation might improve readability. My understanding is that radiative forcings usually sum, such that Eq. 1 could be written as

$$RFari_{allsky} = RFari_{clear} + RFari_{cloudy}$$

rather than a cloud-fraction weighted sum. This would improve readability throughout the paper, as "contribution of cloud sky to RFari" is written much more often than RFari$_{cloudy}$ at the moment (perhaps $RFari_{cloudy} = AC \times RFaci|_{cloudy}$). Having a linear sum of terms would also match better with the approximate linear sum $ERFaer = ERFari + ERFaci$. This is somewhat a matter of taste, so I understand if the authors prefer to stick with the current notation.

Second, while I like the idea of the PCA decomposition, I found it hard to interpret and ended up mostly looking at the correlogram (Fig. 4b). Some more explanation and guidance to interpretation would be useful here. Does it use only the values in Tab. 1 (the global mean values)? What does it mean that SW_CRF has no contribution to PC1, yet has the strongest correlation to the cloudy sky contribution to the RFari in 4b? "Cloudy, FIX2 and FIX3 are plotted but don't affect the projection of the other variables." - I am not quite sure what this means for the interpretation of their position, is this just their correlation with PC1 and PC2? What does this shown. Also, Fig 4c does not appear to be referenced in the text at all. Is this intentional?

Third, how do these value fit in with the "error in the cloud radiative forcing" calculated using the method in Ghan (ACP, 2013)? That method would suggest a contribution to the RFari from aerosol above cloud of +0.40 Wm$^{-2}$. Higher values (although not as large as this) are also found in Gryspeerdt et al. (ACP 2020), which uses essentially the same method.

Finally, a few more commas would be nice to improve readability and there are a few typos which could be caught in the next round (I have identified some of them below)

**Minor points**

L28 - Why are SSA and SW_CRF called out here, when they control PC2?

L68 - substantially

L99 - constraint

L102 - FIX2scat, FIX3abs? These acronyms are used in Tab. 1, but not elsewhere. Having the "scat" and "abs" suffixes is helpful for those less familiar with the experiments.

L121 - Not quite clear what is going on here. Why is the variable for which you are trying to explain the variance added to the list of variables in the PCA?

L151 - "All sky RFari" or RFari$_{allsky}$? I know these are the same, but it might help keep things clear.

L161 - "present-day" instead of "current" would make this clearer that it is not referring to a current estimate.

L184 - SW_CRE vs SW_CRF - Cloud radiative effect is referred to, but the acronym suggests radiative forcing.

L188 - Supplementary information seems to be missing

L189 - FIX2 and FIX3 are hardly used. Is there more that could be said here?

L194 - "PCA finds a weak dependence"

L207 - "However, when analyzing multi-model simulations, additional factors become important."

---

## Author Comment (AC1) · 3 Jun 2020

In this work, the authors investigate the contribution of aerosol in cloudy skies to the magnitude of the aerosol direct effect (RFari). They results from a collection of global models to show that the contribution to the RFari from cloud skies is small. They also investigate the parameters that affect this between different models, showing that the shortwave cloud radiative effect is the biggest contributor to inter-model differences. This work is within scope for ACP and would be relevant for the readers. I have some comments, particularly regarding the notation and some of the explanation, after which I believe it would be suitable for publication.

**Response: we appreciate the nice evaluation and interpretation of our manuscript and constructive comments to improve the manuscript.**

Major points

Has the choice for the meaning of RFariclear and RFaricloudy been made in a previous paper? If not, a change in the notation might improve readability. My understanding is that radiative forcings usually sum, such that Eq. 1 could be written as RF ariallsky = RF ariclear + RF aricloudy rather than a cloud-fraction weighted sum. This would improve readability throughout the paper, as "contribution of cloud sky to RFari" is written much more often than RFaricloudy at the moment (perhaps RF aricloudy = AC × RF aci| cloudy). Having a linear sum of terms would also match better with the approximate linear sum ERF aer = ERF ari + ERF aci. This is somewhat a matter of taste, so I understand if the authors prefer to stick with the current notation.

**Response: We agree to the nice suggestion to improve the readability of the manuscript by simplify Eq 1. We change RFari$_{clear}$ from the earlier version to RFari$_{clear-total}$ and similar the cloudy contribution to continue illustrating the term in Fig 2.**

Second, while I like the idea of the PCA decomposition, I found it hard to interpret and ended up mostly looking at the correlogram (Fig. 4b). Some more explanation and guidance to interpretation would be useful here.

**Response: PCA is a fairly well known, yet complicated, statistical technique used in exploratory data analysis. As such there is limit to how much can be explained in this paper, without making the paper about explaining PCA. What is important to understand is that PCA is a dimension reduction technique, allowing for the visualization of all the variance between the variables in a two-dimensional plot (biplot). When we use a correlogram we only get to see the one to one correlation between variables. With the PCA we can assess the relationship between multiple variables simultaneously. This allows us to get a sense of the degree of influence the variables have on each other. This cannot be seen in a correlogram. However, because the projections of the variables in the biplot are dependent on each other, it can be hard to see the one to one relationship. There is therefore pros and cons to both types of plots, but together they help in the exploratory data analysis.**

**We have included more description of the method and interpretation of the results, see below on response to other comments.**

Does it use only the values in Tab. 1 (the global mean values)?

**Response: Yes, table 1 contains all the data used in the PCA. However, as mentioned on line 122 in the ACPD paper. We have two models, LMDZ-INCA and ECHAM-HAM, which have some missing data. In PCA, no missing data can exist in the analysis. If a single record (i.e. in this case a climate model) is missing data for one variable, then the entire record must be removed for all variables. Removing two models would be fairly detrimental to this study, as we have few records (8 climate models) to begin with. For this reason, we use the technique regularized iterative PCA to fill in the missing values with estimates. As we are only using the PCA to get an overview of the relationship among the variables, and we are not trying to create a predictive model for estimating "Cloudy", the use of this imputation technique should be valid. We have added at the top of the paragraph that global mean used in the Multivariate data analysis.**

What does it mean that SW_CRF has no contribution to PC1, yet has the strongest correlation to the cloudy sky contribution to the RFari in 4b?

**Response: The principle components represent new dimensions created to plot the variance. This is described on line 109-119. If all variables were correlated with PC1, then all the variables would all be correlated with each other. All PC are anticorrelated with each other, which is why PC1 and PC2 are perpendicular to each other. Such is also the case between PC1 and PC3, PC2 and PC3, etc. As plotted in figure 4a) SW_CRE is near perfectly positively correlated with PC2, hence it has to be anticorrelated with every other PC. Which is why the vector is perpendicular to the PC1 axis. This is also why in figure 4c) the length of the vector is so short, as it nether correlates with PC3 or PC1. So yes, SW_CRE is correlated with PC2, and also Cloudy is correlated with PC2. The vector is just pointing in the opposite direction, which means the two variables are negatively correlated with each other. This is also what figure 4b) suggest providing negative correlation between the two variables.**

**We have made the following changes:**

**In the second paragraph in section 2.2**

1) **We have replaced the following sentence: '***Each following PC in turn has the highest variance possible assuming that it is orthogonal to the previous PC, successively explaining less of the magnitude of cloudy sky RFari***' with '***All the variables relationship to each other can be to a lesser degree explained (magnitude) with each exceeding PC. In other words, it's not exclusively to RF ari$_{cloud}$.***'**
2) **This sentence is added: '***All PCs are anticorrelated with each other, which is why PC1 and PC2 are perpendicular to each other.***'**

**In the result section 3.2:**

3) **The following text is added: '***SW_CRE is near perfectly positively correlated with PC2 (Figure 4a), hence it must be anticorrelated with every other PCs. Therefore, the vector in Figure 4a is perpendicular to the PC1 axis. This is also why the length of the vector is so short in Figure 4c since it nether correlates with PC3 or PC1. SW_CRE is correlated with PC2 and Cloudy is correlated with PC2. The vector is pointing in the opposite direction between SW_CRE and Cloudy, which means the two variables are negatively correlated with each other. Figure 4b show also a negative correlation between the two variables.***'**

"Cloudy, FIX2 and FIX3 are plotted but don't affect the projection of the other variables." - I am not quite sure what this means for the interpretation of their position, is this just their correlation with PC1 and PC2? What does this shown.

**Response: In the 4 caption we have added: '**This requirement is made since cloudy, FIX2scat and FIX3abs already depends on the other variable and see their correlation.**'**

Also, Fig 4c does not appear to be referenced in the text at all. Is this intentional?

**Response: It is added that global mean values from Table 1 is used in PCA analysis. Reference to Fig 4c is now included. SW_CRF, FIX2 and FIX3 have been changed to SW_CRE, FIX2scat and FIX3abs, respectively as suggested by the reviewer.**

**We have added the following text in the result section 3.2: '***A biplot with PC1 and PC3 (Figure 4c) can explain more about a variable than PC1 and PC2. For example, CL_ALT has a slightly stronger projection in the PC1 and PC3 biplot and suggest that there is an anticorrelation with FIX2scat. However, in the PC1 and PC2 they are positively correlated with each other. This suggest that there is partial correlation and Figure 4b shows there is a weak positive correlation between these two variables.***'**

**We have added this sentence in section 3.2: '***Adding PC3 this number increases to 89.2%.***'**

Third, how do these value fit in with the "error in the cloud radiative forcing" calculated using the method in Ghan (ACP, 2013)? That method would suggest a contribution to the RFari from aerosol above cloud of +0.40 Wm−2 . Higher values (although not as large as this) are also found in Gryspeerdt et al. (ACP 2020), which uses essentially the same method. Finally, a few more commas would be nice to improve readability and there are a few typos which could be caught in the next round (I have identified some of them below)

**Response: From Table S2 in Gryspeerdt et al. (ACP 2020) the mean SWaricloud is +0.01 W m$^{-2}$ of 8 models. The residual among a large set of simulating the ERFaer (direct and indirect aerosol effects) is weak indicating that SWaricloud is likely weak in all 17 models from AeroCom and CMIP5. We have had the following to the discussion section:**

'*The simulations used in this study only include the RF of the aerosol-radiation interaction. In a recent multi-model study, a decomposition of all aerosol effect (including aerosol-cloud interactions) provides weak RFari$_{cloud}$ for all models, of magnitude and multi-model mean similar to this study (Gryspeerdt et al., 2020). A separate single-model study however found it to be substantial (Ghan, 2013).*'

Minor points L28 - Why are SSA and SW_CRF called out here, when they control PC2?

**Response: These two factors have been shown in this study to play a major in calculation of the RFari$_{cloud}$ and thus emphasized in the abstract.**

L68 - substantially L99 - constraint L102 - FIX2scat, FIX3abs? These acronyms are used in Tab. 1, but not elsewhere. Having the "scat" and "abs" suffixes is helpful for those less familiar with the experiments.

**Response: the comment is taken into account**

L121 - Not quite clear what is going on here. Why is the variable for which you are trying to explain the variance added to the list of variables in the PCA?
**Response: See the additional text added to the manuscript described above.**

L151 - "All sky RFari" or RFariallsky? I know these are the same, but it might help keep things clear.

**Response: Simplifying Eq 1 as suggested by the Reviewer make the readability of this sentence easier and sentence is therefore made much shorter.**

L161 - "present-day" instead of "current" would make this clearer that it is not referring to a current estimate.

**Response: Comment is taken into account.**

L184 - SW_CRE vs SW_CRF - Cloud radiative effect is referred to, but the acronym suggests radiative forcing.

**Response: We agree to the comment and have change SW_CRF to SW_CRE through the manuscript.**

L188 - Supplementary information seems to be missing

**Response: We have changed the reference to Supplementary to Fig 4c.**

L189 - FIX2 and FIX3 are hardly used. Is there more that could be said here?

**Response: We have added the following: 'FIX2scat and FIX3abs are strongly dependent on the host model clouds and their radiative effect and anticorrelated to cloud fraction and SW_CRE, respectively'**

L194 - "PCA finds a weak dependence"

**Response: Sentence corrected to include '*a*' before '*weak*'**

L207 - "However, when analyzing multi-model simulations, additional factors become important."

**Response: Sentence changed as suggested.**

Ghan, S. J.: Technical Note: Estimating aerosol effects on cloud radiative forcing, Atmospheric Chemistry and Physics, 13(19), 9971-9974, 2013.

Gryspeerdt, E., Mülmenstädt, J., Gettelman, A., Malavelle, F. F., Morrison, H., Neubauer, D., Partridge, D. G., Stier, P., Takemura, T., Wang, H., Wang, M. and Zhang, K.: Surprising similarities in model and observational aerosol radiative forcing estimates, Atmos. Chem. Phys., 20(1), 613-623, 2020.

---

## Author Comment (AC2) · 3 Jun 2020

General

The paper presents a modelling study on the direct aerosol effect on climate. The authors distinguish between clear and cloudy skies. The approach is probably state of the art although, however, to my opinion, a very simple one. Let me start with my general impression: We have satellite lidars delivering global 3-D aerosol distributions (profiles!) with detailed aerosol typing (in terms of optical, microphysical and even chemical composition and thus refractive index characteristics) around the globe from the surface up to stratospheric heights and also producing 3-D distributions of clouds layers, their thermodynamic phase, frequency and cloud cover. In addition, we have sophisticated passive remote sensing techniques, again, delivering very detailed information on cloud layering, cloud heights, cloud types, cloud cover, and thermodynamic phase. In view of all the available and complex global 3-D cloud and aerosol data sets, I am a bit surprized that teams of modellers still use rather simple approaches (here Eq.(1)) to investigate and estimate the role of aerosols (natural and anthropogenic ones) in the climate system with the goal to answer the very important and 'ultimate' question: What is the contribution of anthropogenic aerosols to climate change? Even if global MODIS column information on AOD (and maybe cloud occurrence and cover?) is included in the study, . . . is that sufficient to obtain a realistic picture on aerosol effects on climate? The global aerosol distribution (profiles) used in this manuscript is rather simple so that question arises: Does the modelled global aerosol climatology really reflect the real world?

**Response: To the authors it is unsure whether the Reviewer has understood the advanced global modelling and approach applied in this study. It is incorrect that we use Eq 1 to investigate anthropogenic and natural aerosols, our application of Eq 1 is used for anthropogenic aerosols. However, the global modelling includes natural aerosols and simulations of aerosol vertical distribution on a high temporal scale. The Reviewer mention a large set observational data the authors are aware of, but the apparent misunderstanding by the Reviewer is that these observational datasets only provide the present aerosol abundance. E.g. we have already referred to two studies comparing the AeroCom models against CALIPSO in the discussion (Koffi et al., 2016; Koffi et al., 2012). In this study we have the aim of investigate anthropogenic aerosols and the observations cannot provide a clear distinction between anthropogenic and natural aerosols and therefore model information is required. We refer to several studies using observational studies to estimate the cloudy sky RF, and we mention they have large limitations not only on abundance, but also natural and anthropogenic aerosols have large differences in aerosol optical properties. After presenting the earlier studies using observations, we have noted the following: '*Note that the above-mentioned studies investigate the present, total aerosol abundance which consist of anthropogenic and natural aerosols, whereas in terms of RFari only the anthropogenic aerosols are considered*'. In the beginning of the introduction we have now underscored that the estimate of RF is for anthropogenic aerosols. We disagree with the Reviewer that our approach is simple, even though Eq is simple.**

Maybe, there are meanwhile modelling groups and thus papers in which the measured global aerosol distributions and measured global cloud distributions are used to model the impact of aerosols on global climate conditions, and these authors here just want to offer an alternative way, a more simple, rather basic approach to estimate the aerosol effects on climate? Maybe that is the reason for this simple paper but at the end the main question is still: Can we believe in these results when such a simple approach is used?

**Response: see above**

And, are you sure that you cover the full spectrum of anthropogenically caused aerosols. What about all the dust in the atmosphere especially over Central and East Asia, is that all natural? Clearly: NO! But how to consider that in the model? Did you consider that in the simulations? Probably not! The paper is worth to be published, no doubt! The list of authors is full of well-known experts, and the paper is a valuable contribution to the climate debate, but the authors should at least try to provide some answers to my concerns. Yes, maybe I am 'naive' . . . as an experimentally working specialist for aerosol and cloud profiling, and my comments indicate that I am not familiar with the modern modelling world but I am probably not the only one who has trouble with the concept and content of this paper. Maybe, I completely missed the point and the overall message of the paper, but again, I will be probably not the only one. So, we need a more critical discussion on the paper approach itself in this paper.

**Response: We agree that anthropogenic dust aerosols are not included in the models applied in this work and only a very few in current state of the art models. We have therefore added the following sentence in the discussion: 'All the global models that supplied simulations for this study treat the major anthropogenic aerosol components sulphate, organic aerosols, and black carbon, some also treat nitrate, but none include anthropogenic dust aerosols which have highly uncertain radiative effects.'**

Details:

P2, l40: Bellouin et al. . .. this is obviously not a publication, there is no year of publication, nothing. So, that is not an acceptable statement. Please improve!

**Response: This paper is now published, and we have now included a complete reference.**

P2, l50: . . . biofuel BC emission inventory is much higher than used in previous global modelling . . .. Bad wording? What do you want to say?

**Response: Sentence rewritten.**

P2, l62: Eq (1) is the most basic (trivial) approach, right? Or is there even a more simple one? On the other hand, the atmospheric system is so complex, and modern instrumentation fill the aerosol and cloud data base since 20 years, continuously. You seem to ignore all this! You separate (anthropogenic) aerosol particles in absorbing and non-absorbing ones, nothing else. Is that sufficient? You introduce AC as cloud fraction! Obviously it doesn't matter whether we have one layer, two layers, three layers of clouds, whether we have liquid-water clouds, mixed-phase clouds, cirrus . . .or even complex cloud mixtures and layering, and it is also not essential whether the aerosol is below the lowest cloud layer, between the different cloud layers, etc. . . Just one parameter is sufficient: AC! For the entire globe! For rather different climate zones? One AC value everywhere. . .? This is quiet a surprizing and 'universal' assumption. The other way around, what did I miss here? Please clarify, other readers (not familiar with climate modelling) may think the same. . ., may have the same problem with the paper. Maybe all the referenced papers show that it is sufficient to have just AC to describe the impact of clouds on the aerosol radiative effect around the globe from the tropics to the poles.

**Response: All the global models simulate the spatial variation in the vertical profiles of aerosols and clouds and their composition and optical properties. We underscore that the simulations are not only done for a layer, but the study is based on complex global aerosol modelling. See further comments in the response to the main comments. When determining the radiative forcings of**

**RFcloud, RFclear etc, all model grid boxes are utilized, where the vertical distribution of the clouds and different aerosols and their optical properties will affect the radiative forcing calculations.**

P3, l70: aerosols above clouds, below clouds. . . Only these two scenarios, not more are need to be modelled and considered? . . . although the world is full of complex aerosol and cloud layering. . . and large areas over the oceans downwind of polluting continents in the northern hemisphere . . . are 'affected' by this complex layering?

**Response: This is clearly a misunderstanding by the Reviewer, it is certainly not only aerosols above or below the cloud. The models applied in this study have between around 20 to 60 vertical layers in the atmosphere. See further comments above.**

P3,l93: When using Stefan Kinne's aerosol climatology, did you at least check how good the agreement between CALIPSO aerosol profile observations (in combination with MERRA and CAMS simulations) and Kinne's aerosol climatology is? I speculate: Yes, you did that! My 'spontaneous feeling' is that this quiet simple aerosol profile climatology is not in good agreement with the real world. So, please comment on this!

**Response: In the MACv2 climatology the distribution of AOD (where the monthly local statistics of AERONET/MAN corrects the multi-model median of AeroCom phase 1 maps on a regional basis) a distinction is made for AOD from coarse particles (dust, seasalt) and AOD from fine-mode particles (pollution, wildfires). To approximate the aerosol vertical distribution (via vertical scaling of the local monthly column AODc and column AODf data of the MACv2 climatology), scaling factor from 20-year averages from ECHAM5-HAM model simulations are applied. Hereby a single model (ECHAM) was chosen (over a global model median) because in tracer studies and in comparisons to CALIPSO data (paper by Sarah Guibert) this model behaved very well and was not so 'vertical transport-happy' as many other models in that comparison.There was a consideration to replace the vertical aerosol distribution of aerosol with (more observational) data from Calipso. That this has not been included (so far) as CALIPSO data put more aerosol closer to the surface even in comparison to CAMS (e.g. in dust-outflow regions). More importantly CALIPSO data cannot distinguish between fine-mode AOD and coarse-mode AOD. Note than for the study of anthropogenic aerosol only the fine-mode AOD is relevant as anthropogenic aerosol is predominantly an added fraction to the fine-mode AOD. The aerosol vertical distribution also needs to be seen in the context of the cloud altitude placement (The MACv2 climatology distinguishes between high mid and low level clouds, where low clouds are near 1km above the ground, mid-level clouds at ca 3km above the ground). Since random cloud overlap (clouds at 3 altitudes require 8 separate simulations for each permutation) is assumed the cloud-free fraction in MACv2 is on average only at 30%). In MACv2 there is a significant fraction of for optically thin high-only cloud fraction, which may explain a relatively negative forcing for cloudy skies in the comparison. The model description is updated in the manuscript to include information on the vertical profile as follows: *'The Max Planck Aerosol Climatology (MACv2) method combines aerosol column optical properties for fine-mode and coarse-mode sizes (of an AeroCom phase1 model median regionally adjusted by AERONET/MAN monthly statistics) with MODIS surface albedo data, ISCCP cloud properties and vertical scaling by size-mode from 20 years of ECHAM-HAM aerosol simulations. The anthropogenic properties is defined as a fraction of the fine-mode, where the fine-mode AOD scaling factor prescribed from AeroCom phase1 simulations.'*

I would suggest to include a figure with a sketch of your basic aerosol-cloud scenarios considered in the model. Show a cloud layer (provide information on the cloud height, then visualize AC, that means, the cloud should not cover the full sketch from left to right, and then indicate aerosols (just a mixture of black (absorbing) and yellow or white points (non absorbing particles). Scene 1: aerosol

below the cloud, Scene 2: aerosol above the cloud layer, Scene 3: aerosol in the clear part of the sketch, if there are more scenes in the model, please continue with further scenes. . ..

**Response: The model simulations are complex with (multiple grid boxes and) multiple vertical layers with clouds and aerosols of different properties found at different height, all of which varies with time and geographical location. Since the Reviewer has misunderstood that we've just do simulations for one cloud layer (see above and the comment below), we refrain any further response to this comment.**

P5, l127: Result section: My only one question . . . throughout this section. . . was at what height is the cloud layer (for which we have a fixed, constant AC)? Obviously you only consider liquid-water clouds in the lower troposphere. A cloud layer at, e.g., 1 km height (boundary layer top) almost everywhere. . .. around the globe. Maybe it is stated somewhere and I missed it unfortunately. But what about the impact of all the midlevel cloud fields (partly glaciated. . .) and the extended subvisible cirrus fields around the globe. . . , no impact on the aerosol related radiavtive effects?

**Response: Again, the model simulations contain complex treatments of clouds at all altitudes around the global and this reviewer comment is a bit off mark. We have included a sentence in the Result section making it clear that although aerosols are found to have a large effect when located above low clouds, all placement of different aerosols types in relation to cloud are treated in the models, be it above, within or below clouds, for different cloud types (low, mid and high, liquid, mixed and ice).**

The rest of the paper sounds ok (consistent) . . .. for a non-modelling atmospheric scientists traveling around the globe and measuring the rather complex world of clouds and aerosols in regions with very high amounts of haze and dust (which is partly triggered by human activities) and partly complex aerosol layering up to the tropopause, . . . and, in contrast, in very pristine areas with simple cloud and aerosol layering as in your model.

My 'basic' comments may be confusing but the goal is to improve the paper, not to destroy it.

Koffi, B., Schulz, M., Bréon, F.-M., Dentener, F., Steensen, B. M., Griesfeller, J., Winker, D., Balkanski, Y., Bauer, S. E., Bellouin, N., Berntsen, T., Bian, H., Chin, M., Diehl, T., Easter, R., Ghan, S., Hauglustaine, D. A., Iversen, T., Kirkevåg, A., Liu, X., Lohmann, U., Myhre, G., Rasch, P., Seland, Ø., Skeie, R. B., Steenrod, S. D., Stier, P., Tackett, J., Takemura, T., Tsigaridis, K., Vuolo, M. R., Yoon, J. and Zhang, K.: Evaluation of the aerosol vertical distribution in global aerosol models through comparison against CALIOP measurements: AeroCom phase II results, Journal of Geophysical Research: Atmospheres, 121(12), 7254-7283, 2016.

Koffi, B., Schulz, M., Breon, F. M., Griesfeller, J., Winker, D., Balkanski, Y., Bauer, S., Berntsen, T., Chin, M. A., Collins, W. D., Dentener, F., Diehl, T., Easter, R., Ghan, S., Ginoux, P., Gong, S. L., Horowitz, L. W., Iversen, T., Kirkevag, A., Koch, D., Krol, M., Myhre, G., Stier, P. and Takemura, T.: Application of the CALIOP layer product to evaluate the vertical distribution of aerosols estimated by global models: AeroCom phase I results, Journal of Geophysical Research-Atmospheres, 117, D10201, doi:10.1029/2011jd016858, 2012.